bioinformatics/genomics

single-cell sequencing, copy number variation, intra-tumour heterogeneity, overlapping pooling, branch and bound

**Author for correspondence:**
Jing Tu
e-mail: jtu@seu.edu.cn

# Improving the efficiency of single-cell genome sequencing based on overlapping pooling strategy and CNV analysis

## Jing Tu, Zengyan Yang, Na Lu and Zuhong Lu

State Key Laboratory of Bioelectronics, School of Biological Science and Medical Engineering, Southeast University, Nanjing 210096, People's Republic of China

 JT, 0000-0002-1310-5915; NL, 0000-0002-3850-1009; ZL, 0000-0003-3332-2615

Single-cell genome sequencing has become a useful tool in medicine and biology studies. However, an independent library is required for each cell in single-cell genome sequencing, so that the cost grows with the number of cells. In this study, we report a study which efficiently analyses single-cell copy number variation (CNV) using overlapping pooling strategy and branch and bound (*B&B*) algorithm. Single cells were overlapped pooled before sequencing, and later were assorted into specific types by estimating their CNV patterns by *B&B* algorithm. Instead of constructing libraries for each cell, a library is required only for each pool. As the number of pools is smaller than the cells, fewer libraries are required, which means lower cost. Through computer simulations, we overlapped pooled 80 cells into 40 or 27 pools and classified them into cell types based on CNV pattern. The results showed that 84% cells in 40 pools and 76.5% cells in 27 pools were correctly classified on average, while only half or one-third of the sequencing libraries were required. Combining with traditional approaches, our method is expected to significantly improve the efficiency of single-cell genome sequencing.

## 1. Introduction

Genomics variants range from single-nucleotide changes to large chromosomal-level aberrations and can be presented in many forms, including single-nucleotide variations (SNVs), small insertions and deletions (Indels), copy number variations (CNVs), larger structural variants and entire chromosome alteration (Aneuploidy) [1,2]. CNVs originally represent amplifications and deletions of the fragments of genomics DNA which are greater

than 1 kilobase (kb) or less than 5 megabases (Mb) in size [3,4]. With the development of methodologies, the concept of CNVs is later widened where smaller (50 b–1 kb) amplifications and deletions are included [5]. More and more studies have shown that CNVs confer susceptibility to a variety of human cancers [6,7]. Inter-tumour and intra-tumour heterogeneity pose a great challenge for cancer treatment and recovery [8,9].

With the emergence of whole-genome amplification methods, single-cell sequencing technologies are able to reveal the heterogeneity of cells within tumours in CNVs [10,11]. However, present studies often analyse a small number of single cells (dozens to hundreds) in one time because of cost and technological limitations [12,13]. It is still a challenge to evaluate copy number heterogeneity for a large amount of cells. A more efficient single-cell sequencing approach is desired to analyse large-scale single cells. In fact, the main purpose for single-cell CNV analysis is to cluster specific cells into subclones, which are the subpopulations of cells with distinct genotypes [10]. Therefore, by characterizing the subclones in prior, basically accurate CNV patterns are adequate for the cells to be correctly classified. The subpopulation of cells can be divided by sequencing a part of single cells in prior, and the CNV patterns of subclones can be obtained at the same time. Classifying more single cells into subclones is a benefit to comprehensively reveal the heterogeneity of the sample, especially when these cells contain additional information, such as clinical information, pathological information or spatial information.

Based on this premise, we propose a novel strategy which is expected to greatly reduce the costs of single-cell sequencing by using overlapping pooling method and branch and bound (*B&B*) algorithm to assort single cells into subclones with fewer sequencing libraries. In our strategy, several pools are built, and single cells are overlapped and put into these pools randomly. Each pool is sequenced independently, and their CNV patterns are identified. By calculating two ingenious equations, the CNV patterns of each single cell can be classified into the known types. This novel strategy was proof-of-concept studied here through implementing computer simulations on datasets with a medium cell number. In total, 80 single cells were placed into 40 or 27 pools, and their CNV patterns were further classified. The results showed that 84% cells in 40 pools and 76.5% in 27 pools could be correctly classified to their origin CNV patterns on average. Compared with the traditional methods, only a half or one-third of sequencing libraries are required in our method.

# 2. Material and methods

## 2.1. Resources

The human reference genome sequence (hg19) was downloaded from the University of California, Santa Cruz (UCSC) Genome Browser database [14]. The single-cell sequencing data from seven tumour patients with triple-negative breast cancer (TNBC) were downloaded in FASTQ format from National Center for Biotechnology Information (NCBI) [15] under sequence read archive (SRA) accessions SRP064210 [7]. We randomly selected multi-group single-cell sequencing data (36/80/137 single cells) for testing. Finally, sequencing data of 80 single cells from seven tumour patients with TNBC were used in this paper.

## 2.2. Sequence alignment and data processing

We followed the protocol put forward by Baslan *et al.* [16] to obtain the copy number profile of single cells. In brief, firstly, we mapped qualified short-sequencing reads to hg19 by using the Burrows–Wheeler Aligner (BWA, 0.7.15-r1142-dirty) [17] with maximal exact matches algorithm and defined parameters to generate SAM files. Once mapped, sequencing reads are processed using the SAMtools (1.8) package [18] to convert the SAM files to compressed BAM files and sort the BAM files in a proper format suitable (followed by chromosome coordinates) for downstream analysis. To remove PCR duplicates, the Picard (1.141) [19] 'MarkDuplicates' was used to remove short reads with identical start coordinates from sorted BAM files. Short-sequence reads with low-mapping quality (less than 40) were also filtered out from de-duplicated BAM files using Sambamba [20].

## 2.3. Copy number profiles calculation

Only uniquely mapped reads were then used in counting the number of sequencing reads in 28 416 genomic intervals (bins) with an average length of 100 kb. The single cells with greater than or equal

to 50 median reads/bin were selected for calculating copy number. Then, a locally weighted scatterplot smoothing (LOESS) regression model [16] was applied for performing estimation and correction of guanine-cytosine (GC)-content bias. We segmented copy number profiles with circular binary segmentation and merged adjacent segments that were not significantly distinct in segmented log2-ratios using MergeLevels. Finally, the integer copy number was calculated by scaling segmented log2-ratios with average DNA ploidy and rounding to the closest integer values. It should be noted that all the above processes were executed under default parameters.

## 2.4. Pooling pattern matrix designation

The pooling pattern matrix $PC$ was designed at first, which is a $j \times t$ binary matrix. The rows are indexed by pools $P1, \ldots, Pt$ and columns are indexed by samples (cells) $C1, \ldots, Ct$. In this matrix, $PCij = 1$ if and only if the $j$th cell (sample) is contained in the $i$th pool. Neither the sum of each row nor the column is 0. The reversion is based on two equations and five matrixes.

Before starting solving these two equations, the CNV pattern of each type should be obtained. First, all cells were classified using hierarchical clustering analysis based on CNV patterns. Next, to obtain typical CNV bins of each cell, we counted the distribution of copy number of all samples on every bin, and calculated the frequency of each copy number, then discarded bins where the maximum frequency of copy number is lower than thresholds (0.8, the detailed introduction presented in §3.3) and kept the rest. Cells sharing the same copy numbers were combined into type. Then, we discarded bins where all types are 2 (normal copy number, diploid) and got the matrix $TB$. After sequencing pools, analysing data and calculating the copy number profiles of each pool, we got the copy number of each bin. Bins which were discarded in the construction of matrix $TB$ were also discarded here and the matrix of reserved bins was name matrix $PB$.

# 3. Results

## 3.1. The principle of overlapping pooling single-cell sequencing

To solve the high cost of library construction in single-cell sequencing, we introduced the pool strategy to get satisfactory results. Pooling cells directly cannot meet up with the goal of reducing libraries while acquiring sufficient information in single cells, so that we added overlapping strategy to build relevance among different pools. In this novel strategy, cells are randomly overlapped pooling into a series of pools, where the number of pools is less than the number of single cells. Instead of constructing libraries for each single cell, whole-genome sequencing library is constructed for each pool which is sequenced subsequently. Due to the number of pools being smaller than that of the single cells, if we could obtain the type of each original single cell from the sequencing results of pools, parts of the libraries are cut, lower cost is required, and the efficiency of single-cell genome sequencing is improved.

As shown in figure 1, the estimation of cell type (which CNV pattern of cell's CNV belongs to, one cell one pattern) can be transformed into solving the matrix $CT$ (cell type) based on two equations and five matrixes. In brief, the matrix $PT$ (pool-type) can be solved based on CNV results of each pool (pool-bin matrix, matrix $PB$) which is got by sequencing pools, calculating the copy number profiles of each pool and the pre-known characters of cell type (type-bin matrix, matrix $TB$). After that, we try to find an optimal matrix that has the smallest gaps from the exact solution for the underdetermined matrix $CT$ using $B\&B$ algorithm based on the designed pooling pattern matrix (pool-cell matrix, matrix $PC$) and the solved matrix $PT$. The solving of pooling pattern matrix is described in detail in the following.

## 3.2. Pooling pattern matrix solution

With matrices $PB$ and $TB$, we could solve the first equation and get the fourth matrix $PT$. This system has more equations than unknowns and it is obviously not square so that it does not have an inverse matrix. We calculated the matrix $TB$'s generalized inverse matrix $TB^g$ which makes up the drawback of regular inverse. Then, the matrix $TB^g$ was multiplied with the matrix $PB$ to get the matrix $PT$ as answer. This matrix $PT$ is a least-squares solution that explains the proportion of different CNV types in each pool.

In solving the second equation, the matrix $CT$ is underdetermined and every row only has one '1' and others are '0' (formula (3.1)). In this case, our goal is to find an optimal matrix that has the smallest gaps

equation 1

$$\begin{array}{c} \\ \text{pool1} \\ \text{pool2} \end{array}\begin{bmatrix} \text{type1} & \text{type2} & \text{type3} \\ 1 & 1 & 0 \\ 0 & 2 & 1 \end{bmatrix} * \begin{array}{c} \\ \text{type1} \\ \text{type2} \\ \text{type3} \end{array}\begin{bmatrix} \text{bin1} & \text{bin2} & \text{bin3} & \text{bin4} & \text{bin5} \\ 2 & 2 & 3 & 3 & 4 \\ 4 & 4 & 3 & 5 & 2 \\ 1 & 4 & 6 & 3 & 3 \end{bmatrix} = \begin{array}{c} \\ \text{pool1} \\ \text{pool2} \end{array}\begin{bmatrix} \text{bin1} & \text{bin2} & \text{bin3} & \text{bin4} & \text{bin5} \\ 3 & 3 & 3 & 4 & 3 \\ 3 & 4 & 4 & 4 & 2 \end{bmatrix}$$

④ $PT$  　　　② $TB$  　　　③ $PB$

equation 2

$$\begin{array}{c} \\ \text{pool1} \\ \text{pool2} \end{array}\begin{bmatrix} \text{cell1} & \text{cell2} & \text{cell3} & \text{cell4} \\ 1 & 1 & 0 & 0 \\ 0 & 1 & 1 & 1 \end{bmatrix} * \begin{array}{c} \\ \text{cell1} \\ \text{cell2} \\ \text{cell3} \\ \text{cell4} \end{array}\begin{bmatrix} \text{type1} & \text{type2} & \text{type3} \\ 1 & 0 & 0 \\ 0 & 1 & 0 \\ 0 & 1 & 0 \\ 0 & 0 & 1 \end{bmatrix} = \begin{array}{c} \\ \text{pool1} \\ \text{pool2} \end{array}\begin{bmatrix} \text{type1} & \text{type2} & \text{type3} \\ 3 & 3 & 3 \\ 3 & 4 & 4 \end{bmatrix}$$

① $PC$  　　　⑤ $CT$  　　　④ $PT$

**Figure 1.** Two matrix equations designed for pooling. The numbers in the equations are examples for matrix calculations and can be replaced with any numbers. The matrix $PC$ represents the distribution of cells in each pool, and its value is known. The matrices $TB$ and $PB$ are calculated from the copy numbers of samples. The matrices $PT$ and $CT$ are calculated by equation 1 and 2. Pool represents the number of pools used in here. Bin represents the genome intervals, the size of bins is variable, and average length is 100 kb. Cell represents the label of cells used in here. Type represents CNV patterns.

from the exact solution (formula (3.2)). Thus, we transformed the equation solution into a linear programming model. Then, *B&B* algorithm was performed with 100 passes at the root node of the *B&B* tree and two passes at all subsequent nodes. Finally, we got the solution matrix $CT$, which reflects to which type a cell belongs. We can rebuild the CNV pattern of each cell as we already know that of each type.

$$\forall_j \sum_{t=1}^{\text{types}} CT(j,t) = 1 \tag{3.1}$$

and

$$\min = \sum_{t=1}^{\text{types}} \sum_{i=1}^{\text{pools}} \left[ \sum_{j=1}^{\text{cells}} PC(i,j) * CT(j,t) - PT(i,t) \right]^2. \tag{3.2}$$

## 3.3. Simulation of overlapping pooling single-cell sequencing

Before we start solving these two equations, the 80 cells were classified into seven types with hierarchical clustering analysis by their CNV patterns (figure 2). When determining the appropriate threshold to find typical CNV bins of each cell, we tried 0.75, 0.8 and 0.85 as threshold. The origin 28 416 bins remained 12 670, 10 681 and 5673, respectively. Considering the solutions' quality and proportion, we chose a proportion of modes higher than 0.8 as the threshold. Then, we discarded bins whose copy number was diploid and got the matrix $TB$.

Based on the designed matrix $PC$, cells in the same pools were merged by extracting 1 million reads of each cell randomly from their BAM files to ensure they are of the same weight. The depths of each bin in the newly generated pools were calculated. Bins which were discarded in the construction of matrix $TB$ were also discarded here and the reserved bins were used to form matrix $PB$.

To demonstrate the precision of our methods, we performed a series of experiments with different combinations of parameters repeatedly. An evaluation standard based on the restoration ratio to score was also established. The score were affected by many factors in the results. As for the first designation of $PC$ matrix, there are three parameters: the number of cells ($c$), the number of pools ($p$)

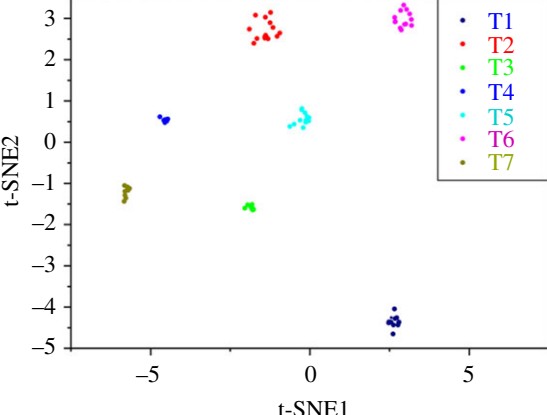

**Figure 2.** Two-dimensional tSNE representation of all single cells included in the study ($n = 80$). Hierarchical clustering of all single cells on the basis of their CNV patterns. Cell clusters are differentially coloured and identified as distinct TNBC patients. All single cells were divided into seven distinct TNBC patients by hierarchical clustering.

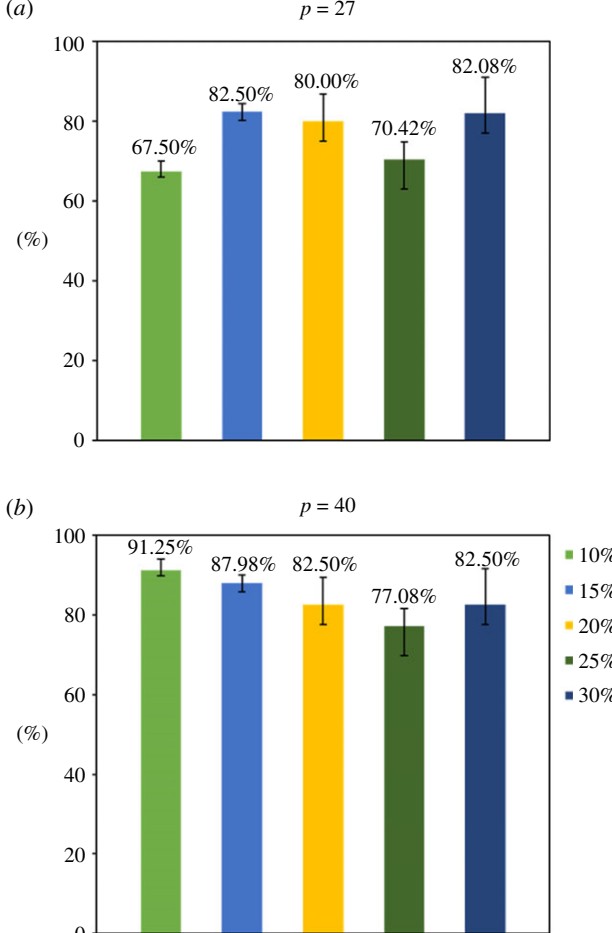

**Figure 3.** The percentage of correctly restored cells under different pools number $p$ and cells per cent $k$. $k \times c$ means the number of cells put into all pools. The percentage values represent the average of three repeated experiments.

and the per cent of cells ($k$). $c$ was determined at first as 80. For $p$, we tried 40 (one half) and 27 (one third) pools, where the former are obviously better than the latter. As for $k$, we attempted five tests of 10% (8), 15% (12), 20% (16), 25% (20) and 30% (24) with every pool number and each repeated three times. The results showed that 84% cells in 40 pools and 76.5% cells in 27 pools could be accurately restored to their original CNV types (figure 3). When the per cent of cells $k$ was set as 10%, a significant difference can be observed between $p = 27$ and $p = 40$.

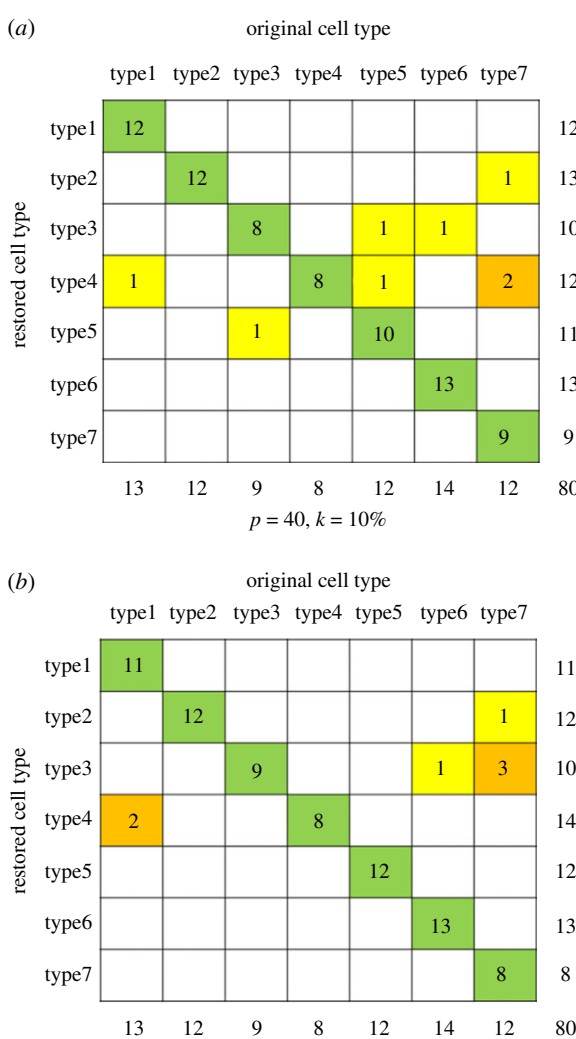

**Figure 4.** Restored cells type with different percentage of cells.

Take $p = 40$, $k = 10\%$ and $k = 30\%$ for an example (figure 4), apart from both of their restoration were of high accuracy (91.25% and 82.50% on average), we can find some similarities and differences between them: type2 was restored entirely but in the clustering, it was the most dispersive type (figure 2), while type7 and type1 seemed the most intensive types (figure 2), were the two worst in restorations. Also, we can find from the figure 4 that two and three cells of type7 were restored to type4, one cell from type1 was frequently divided into type4.

# 4. Discussion

Single-cell genome sequencing technologies are powerful methods to dissert intra-tumour genetic heterogeneity at single-cell resolution. Resolving clonal substructure and reconstructing phylogenetic lineages are the way to investigate clonal evolution in tumour [21]. It has been demonstrated that most CNVs were acquired before SNVs in the evolution of tumour [22], which means a higher resolution in describing the genetic heterogeneity in tumour. However, constrained to the research funding, it is still a challenge to investigate clonal evolution from a large number of cells. In fact, although the genomic diversity varies among tumours and samples, most studies divided cells within a tumour into a limited number of subpopulations to improve the effectiveness of the analysis. For example, Yu *et al.* [23] identified two subsets in the cell population of colon cancer, Gao *et al.* [7] identified one–three major subpopulations of clones in each tumour of the patient with TNBC, and Leung *et al.* [24] identified three–four major cell clusters in the sample of metastatic colorectal cancer. The limited number of subpopulations with distinct genotypes can be identified by sequencing

dozens to hundreds of cells. For more cells, the CNV pattern does not need to be high-accurately detected as long as cells can be correctly classified into specific subclones. If a large number of cells containing different clinical information were classified, the intra-tumour heterogeneity and the evolution of tumour will be revealed more distinctly. The strategy we conceived here provides a convenient and effective way to sequencing a large number of single cells at affordable cost. According to our blueprint, assuming there are 10 000 cells from a cancer tissue needing to be sequenced and classified to study the evolution of tumour, the overall composition of cells and the CNV patterns of subclones can be acquired by randomly sequencing 1000 cells one by one, and the other 9000 cells can be assorted to subclones by conducting our strategy. This has been demonstrated in simulation; only 4500 (a half) or 3000 (one third) sequencing libraries are required for the rest of the cells. More than 4500 sequencing libraries are cut down in comparing with directly sequencing strategy.

Although our method showed a noticeable advantage in reducing library number of single-cell DNA sequencing while maintaining CNV pattern accurately, care must be taken in generalizing our approach as the composition of cell population varies in different samples. There are some issues that should be noticed on the population composition of single cells before the strategy is applied. The CNV patterns of different subclones are desired to be distinct among each other, and cells belonging to the same subclones are desired to obtain highly similar CNV patterns. Subpopulations with a significant difference in CNV pattern means their Euclidean distances are of different orders of magnitude, and cells highly similar in CNV pattern means their Euclidean distances are of same orders of magnitude, which are both benefits to get good performance in cell assorting. Meanwhile, the little difference between cells in the same subclone will be ignored, which will lead to the deviation from the true copy number. The cells with rare CNV pattern will also be ignored and incorrectly classify into one of the cell types.

Besides, the increase of pool number ($p$) showed a noticeable improvement of restoration accuracy, but the effect of cell per cent ($k$) is not clear while the results showed more or less random. When the pool number was large ($p = 40$), the tests of a smaller percentage of cell showed better restoration results. It is probably due to more cells in a pool getting highly complex CNV results, which might get more bias in calculating the approximate solution. However, when the pool number was 27, the tests of $k = 10\%$ got the worst restoration accuracy. The reason is probably that the information gathered in the tests of 27 pools × 10% cells was not enough to support full reconstruction of the initial matrix. It seems that enough information was gathered in the tests of 27 pools × 15% cells, as the restoration accuracy was much better. Further studies are desired to comprehensively discover the contribution of parameters, $p$, $c$ and $k$, towards the accuracy of solutions.

# 5. Conclusion

In summary, we reported a proof-of-concept study of a novel single-cell sequencing approach, which assorts cells into subclones using overlapping pooling method and *B&B* algorithm. Without constructing libraries for each cell, only a half or less sequencing libraries are required to assort the cells into subclones. The overall cost is significantly saved considering the library construction cost as one of the main factors in single-cell DNA studies. Our method is expected to significantly improve the efficiency of single-cell DNA sequencing after combining with traditional approaches.

Data accessibility. The original single-cell sequencing data of 80 single cells from seven tumour patients were downloaded and available online in NCBI under SRA accessions SRP064210. Sequencing data statistics files and copy number profiles of each single cell were deposited in the Dryad Data Repository: https://doi.org/10.5061/dryad.v6wwpzgwr.
Authors' contributions. J.T.: conceptualization, funding acquisition, methodology, supervision and writing—review and editing; Z.Y.: data curation and software; N.L.: data curation, resources and software; Z.L.: methodology and supervision. All authors gave final approval for publication and agreed to be held accountable for the work performed therein.
Competing interests. The authors declared that they have no conflicts of interest to this work.
Funding. This work was supported by National Natural Science Foundation of China (grant no. 61971125) and Six Talent Peaks Project of Jiangsu Province (grant no. 2019-SWYY-004).

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
