## [Peer Review File · Royal Society Open Science]

Review History

RSOS-211330.R0 (Original submission)

Review form: Reviewer 1

Is the manuscript scientifically sound in its present form?

No

Are the interpretations and conclusions justified by the results?

Yes

Is the language acceptable?

Yes

Do you have any ethical concerns with this paper?

No

Have you any concerns about statistical analyses in this paper?

No

Recommendation?

Major revision is needed (please make suggestions in comments)

Comments to the Author(s)

The authors submitted a manuscript reporting a study assorting cells into subclones using the overlapping pooling method without constructing libraries for each cell. The aim of the study is novel, however, it looks the results did not fully support the conclusions.

Here are my comments:

1. In the simulation, the legitimacy of using 80 cells should be stated. Is that the conventional number of cells used in any single cell genome study?
2. Could bootstrap be performed with more repeats in your simulation of the pooling library? It is unclear whether reads in the pooling libraries from the actual experiment are in the same pattern as simulated?
3. Results should be succinct and simple. There is too much jargon. Section 4.2 is like method
4. Legend of figure 2 should be more descriptive.

Review form: Reviewer 2

Is the manuscript scientifically sound in its present form?

Yes

Are the interpretations and conclusions justified by the results?

Yes

Is the language acceptable?

Yes

Do you have any ethical concerns with this paper?

No

Have you any concerns about statistical analyses in this paper?

No

Recommendation?

Accept with minor revision (please list in comments)

Comments to the Author(s)

The authors employed branch and bound (B&B) algorithm to analyze CNV in the in silico overlapping pooling single cell sequencing data. In general, the study is well designed and performed. However, the following concerns need to be addressed.

1. The title of the manuscript is not specific and could mislead readers. The study was centered in CNV analysis of single cell sequencing. Suggest to include CNV analysis in the title.
2. Language need to polished. Some sentences are difficult to read and confusing. Grammar errors present.
3. Pooling parameter k seems affect the restoration rate, however it was not discussed.
4. A significant discrepancy of restoration rate has been observed on figure 3 between $p=40$ and $p=27$ when $k=10\%$. Readers might be interested in more discussion.

Decision letter (RSOS-211330.R0)

Dear Dr Tu

On behalf of the Editors, we are pleased to inform you that your Manuscript RSOS-211330 "Improving the efficiency of single cell genome sequencing based on overlapping pooling strategy" has been accepted for publication in Royal Society Open Science subject to minor revision in accordance with the referees' reports. Please find the referees' comments along with any feedback from the Editors below my signature.

We invite you to respond to the comments and revise your manuscript. It will be important to address comprehensively all of the comments of both reviewers. Below the referees' and Editors' comments (where applicable) we provide additional requirements. Final acceptance of your manuscript is dependent on these requirements being met. We provide guidance below to help you prepare your revision.

Please submit your revised manuscript and required files (see below) no later than 7 days from today's (ie 29-Nov-2021) date. Note: the ScholarOne system will 'lock' if submission of the revision is attempted 7 or more days after the deadline. If you do not think you will be able to meet this deadline please contact the editorial office immediately.

on behalf of Dr Paul Schofield (Associate Editor) and Steve Brown (Subject Editor)
openscience@royalsociety.org

Associate Editor Comments to Author (Dr Paul Schofield):

Associate Editor: 1

Comments to the Author:

The referees have suggested several issues that require some further comments or discussion, and while neither indicated that more work needs to be done, these comments should be addressed in the resubmitted manuscript. Of more concern is the comment from both referees that the text is somewhat opaque and difficult to follow in many places. We request that you address these concerns carefully and in particular pay attention to terms, acronyms and algorithms that might be familiar to those working in precisely this area but not more general readers. The paper should

be understandable by a general reader with an interest in the subject without recourse to looking up citations to understand the text of the manuscript. Overall the language in the paper should be looked at carefully for grammar and expression.

Reviewer comments to Author:

Reviewer: 1

Comments to the Author(s)

The authors submitted a manuscript reporting a study assorting cells into subclones using the overlapping pooling method without constructing libraries for each cell. The aim of the study is novel, however, it looks the results did not fully support the conclusions.

Here are my comments:

1. In the simulation, the legitimacy of using 80 cells should be stated. Is that the conventional number of cells used in any single cell genome study?
2. Could bootstrap be performed with more repeats in your simulation of the pooling library? It is unclear whether reads in the pooling libraries from the actual experiment are in the same pattern as simulated?
3. Results should be succinct and simple. There is too much jargon. Section 4.2 is like method
4. Legend of figure 2 should be more descriptive.

Reviewer: 2

Comments to the Author(s)

The authors employed branch and bound (B&B) algorithm to analyze CNV in the in silico overlapping pooling single cell sequencing data. In general, the study is well designed and performed. However, the following concerns need to be addressed.

1. The title of the manuscript is not specific and could mislead readers. The study was centered in CNV analysis of single cell sequencing. Suggest to include CNV analysis in the title.
2. Language need to polished. Some sentences are difficult to read and confusing. Grammar errors present.
3. Pooling parameter k seems affect the restoration rate, however it was not discussed.
4. A significant discrepancy of restoration rate has been observed on figure 3 between $p=40$ and $p=27$ when $k=10\%$. Readers might be interested in more discussion.

===PREPARING YOUR MANUSCRIPT===

one version should clearly identify all the changes that have been made (for instance, in coloured highlight, in bold text, or tracked changes);

===PREPARING YOUR REVISION IN SCHOLARONE===

-- Ensure that your data access statement meets the requirements at <https://royalsociety.org/journals/authors/author-guidelines/#data>. You should ensure that you cite the dataset in your reference list. If you have deposited data etc in the Dryad repository, please only include the 'For publication' link at this stage. You should remove the 'For review' link.

-- If you are requesting an article processing charge waiver, you must select the relevant waiver option (if requesting a discretionary waiver, the form should have been uploaded, see 'File upload' above).

-- If you have uploaded any electronic supplementary (ESM) files, please ensure you follow the guidance at <https://royalsociety.org/journals/authors/author-guidelines/#supplementary-material> to include a suitable title and informative caption. An example of appropriate titling and captioning may be found at https://figshare.com/articles/Table_S2_from_Is_there_a_trade-off_between_peak_performance_and_performance_breadth_across_temperatures_for_aerobic_scope_in_teleost_fishes_/3843624.

Author's Response to Decision Letter for (RSOS-211330.R0)

See Appendix A.

RSOS-211330.R1 (Revision)

Review form: Reviewer 1

Is the manuscript scientifically sound in its present form?

No

Are the interpretations and conclusions justified by the results?

Yes

Is the language acceptable?

Yes

Do you have any ethical concerns with this paper?

No

Have you any concerns about statistical analyses in this paper?

Yes

Recommendation?

Accept as is

Comments to the Author(s)

The manuscript is improved

Review form: Reviewer 2

Is the manuscript scientifically sound in its present form?

Yes

Are the interpretations and conclusions justified by the results?

Yes

Is the language acceptable?

Yes

Do you have any ethical concerns with this paper?

No

Have you any concerns about statistical analyses in this paper?

No

Recommendation?

Accept as is

Comments to the Author(s)

My comments on the original submission has been addressed. No additional comments. I recommend acceptance of this manuscript.

Decision letter (RSOS-211330.R1)

Dear Dr Tu,

It is a pleasure to accept your manuscript entitled "Improving the efficiency of single cell genome sequencing based on overlapping pooling strategy and CNV analysis" in its current form for publication in Royal Society Open Science. The comments of the reviewer(s) who reviewed your manuscript are included at the foot of this letter.

on behalf of Dr Paul Schofield (Associate Editor) and Steve Brown (Subject Editor)
openscience@royalsociety.org

Reviewer comments to Author:

Reviewer: 2

Comments to the Author(s)

My comments on the original submission has been addressed. No additional comments. I recommend acceptance of this manuscript.

Reviewer: 1

Comments to the Author(s)

The manuscript is improved

Follow Royal Society Publishing on Twitter: [@RSocPublishing](https://twitter.com/RSocPublishing)

Appendix A

Associate Editor Comments to Author (Dr Paul Schofield):

Associate Editor: 1

Comments to the Author:

The referees have suggested several issues that require some further comments or discussion, and while neither indicated that more work needs to be done, these comments should be addressed in the resubmitted manuscript. Of more concern is the comment from both referees that the text is somewhat opaque and difficult to follow in many places. We request that you address these concerns carefully and in particular pay attention to terms, acronyms and algorithms that might be familiar to those working in precisely this area but not more general readers. The paper should be understandable by a general reader with an interest in the subject without recourse to looking up citations to understand the text of the manuscript. Overall the language in the paper should be looked at carefully for grammar and expression.

Response: Thanks for your comments. We revised our manuscript according to your comments and reviewers' comments. Point-by-point response to the reviewers' comment is listed below. We have done our best to make the pipeline clear and easy to follow. Meanwhile, the language has been polished carefully for grammar and expression.

Reviewer: 1

Comments to the Author(s)

The authors submitted a manuscript reporting a study assorting cells into subclones using the overlapping pooling method without constructing libraries for each cell. The aim of the study is novel; however, it looks the results did not fully support the conclusions.

Here are my comments:

1. In the simulation, the legitimacy of using 80 cells should be stated. Is that the conventional number of cells used in any single cell genome study?

Response: Thanks. Actually, the number of single cells used in single cell genome study is uncertain, for example, 47 single cells (10.1093/bib/bbx004), 100 single cells (10.1038/nprot.2012.039), 900 single cells (10.1016/j.cell.2018.03.041) and over 10 thousand single cells (10.1038/nmeth.4154). In addition, we downloaded all single cell sequencing data of the 7 tumor patients from this work (10.1038/ng.3641). Then, 40 cells, 80 cells and 140 cells were randomly selected from the downloaded single cells. Considering the analysis efficiency and the accuracy of result, finally, we used 80 single cells for showing in this paper.

2. Could bootstrap be performed with more repeats in your simulation of the pooling library? It is unclear whether reads in the pooling libraries from the actual experiment are in the same pattern as simulated?

Response: Thanks. First, all the single cell sequencing data used in this paper were downloaded from the articles published on *Nature Genetics*. The sequencing data is reliable, and the number of reads in the sequencing data of each single cell is different. Second, we have also made settings for the number of input single cell and have done pool simulation for different single cell number. This is a proof-of-concept study, so we did not perform specific experiments to verify this method. But our simulation intersects specific experiments, which can get the real authenticity and accuracy of the results.

3. Results should be succinct and simple. There is too much jargon. Section 4.2 is like method

Response: Thanks for your reminding, we moved the part of “Pooling pattern matrix designation” to method section. And the part of “Pooling pattern matrix solution” belongs to the result part, so we stayed in place.

4. Legend of figure 2 should be more descriptive.

Response: Thanks for your reminding, we added more descriptive and clearer explanation of the Figure 2 as following. 2D-tSNE representation of all single cells included in the study (n = 80). Hierarchical clustering of all single cells on the basis of their CNV patterns. Cell clusters are differentially colored and identified as distinct TNBC patients. All single cells were divided into 7 distinct TNBC patients by hierarchical clustering.

Reviewer: 2

Comments to the Author(s)

The authors employed branch and bound (B&B) algorithm to analyze CNV in the in silico overlapping pooling single cell sequencing data. In general, the study is well designed and performed. However, the following concerns need to be addressed.

1. The title of the manuscript is not specific and could mislead readers. The study was centered in CNV analysis of single cell sequencing. Suggest to include CNV analysis in the title.

Response: Thanks for your reminding, we have added CNV analysis to our title which is modified to “Improving the efficiency of single cell genome sequencing based on overlapping pooling strategy and CNV analysis” now.

2. Language need to polished. Some sentences are difficult to read and confusing. Grammar errors present.

Response: Thanks. We polished our language and made some modifications to the sentences.

3. Pooling paramant k seems affect the restoration rate, however it was not discussed.

4. A significant discrepancy of restoration rate has been observed on figure 3 between $p=40$ and $p=27$ when $k=10\%$. Readers might be interested in more discussion.

Response: Thanks for your comments. As the two comments are all concern with the cell percent k , we response them together. We added the results and discussion of the pooling parameter k . When the pool number was large ($p = 40$), the tests of smaller percentage of cell showed better restoration results. It is probably due to that more cells in a pool gets highly complex CNV results which might get more bias in calculating the approximate solution. However, when the pool number was 27, the tests of $k = 10\%$ got the worst restoration accuracy. The reason is probably due to that the information gathered in the tests of 27 pools \times 10% cells were not enough to support fully reconstructed for the initial matrix. It seems that enough information was gathered in the tests of 27 pools \times 15% cells as the restoration accuracy was much better. Meanwhile, an overall higher restoration rate be found in $p=40$ pools as there are more information provided. As for different k (from 0.1 to 0.25) within the same $p=40$, we could see a negative correlation between restoration rate of cells and k . In our opinion, with the number of cells increased in a pool, CNV patterns among the cells conflict with each other so that the restoration is obscured. But when $k=30\%$, the preserved interesting biological structure trumps the influence of the chaos. Due to the feasibility in practical applications, we did not make more attempts for larger k .